# Pregelatinized green banana flour snack bars: Physicochemical properties, sensory quality, and gastroprotective potential in an ethanol-induced ulcer model

Thammarat Kaewmanee[1], Surasak Limsuwan[2], Sineenart Sanpinit[3], Pinanong Na-Phatthalung[4], Samuel Abiodun Kehinde[5], Acharaporn Issuriya[6]*, Sasitorn Chusri[5]*

1 Department of Food Science and Nutrition, Faculty of Science and Technology, Prince of Songkla University, Pattani, Thailand, 2 Traditional Thai Medical Research and Innovation Center, Faculty of Traditional Thai Medicine, Prince of Songkla University, Songkhla, Thailand, 3 Department of Applied Thai Traditional Medicine, School of Medicine, Walailak University, Nakhon Si Thammarat, Thailand, 4 Division of Hematology and Oncology, Icahn School of Medicine at Mount Sinai, New York, New York, United States of America, 5 School of Health Science and Biomedical Technology Research Group for Vulnerable Populations, Mae Fah Luang University, Chiang Rai, Thailand, 6 Division of Health and Applied Sciences, Faculty of Science, Prince of Songkla University, Songkhla, Thailand

☯ These authors contributed equally to this work.
* sasitorn.chu@mfu.ac.th (CS); acharaporn.i@psu.ac.th (IA)

## Abstract

Peptic ulcers remain a recurrent gastrointestinal disorder requiring preventive nutritional strategies. Functional foods that promote gastrointestinal health may play an important role in preventing and managing peptic ulcers. In this study, we evaluated pregelatinized green banana flour (GBF) prepared by drum drying as a functional ingredient for developing snack bars with gastroprotective potential. The aim of this study was to systematically investigate how drum-drying pregelatinization influences the physicochemical and gastroprotective properties of GBF and to determine its suitability as a functional ingredient in the formulation of shelf-stable, consumer-acceptable snack bars. The pregelatinization process significantly reduced the moisture content and water activity, while improving water solubility index and swelling power. Snack bars formulated with 40% pregelatinized GBF (sPreF3) showed the most desirable sensory profile, achieving high hedonic scores for crunchiness (7.470 ± 0.860) and overall appearance (7.40 ± 0.93) on a 9-point scale. During six weeks of storage at 25 °C and 50–70% relative humidity, the product maintained stable moisture content (3.190–4.150%), low microbial counts (< $1.2 \times 10^2$ CFU/g), and consistent sensory acceptability. In vivo, both native GBF and pregelatinized GBF (1,000 mg/kg) significantly reduced ethanol-induced gastric ulcer area (0.020 ± 0.010 cm² and 0.090 ± 0.030 cm², respectively) compared with vehicle controls (0.410 ± 0.090 cm²; $p < 0.05$), while increasing gastric mucin levels (26–27 μg/g tissue) and pH (4.5–5.0). This finding indicates that pregelatinized GBF exhibits

**Data availability statement:** All relevant data are within the manuscript and its Supporting Information files.

**Funding:** The research received financial support from the National Science, Research, and Innovation Fund (NSRF) and Mae Fah Luang University (Fundamental Fund Grant No. 662A05032). It was also partly supported by the Southern Thailand Science Park (Collaborative Research Program; Co-Research-PsuCor-1/62), Prince of Songkla University. The manuscript's production was partially assisted through academic collaboration under the reinventing university system of Mae Fah Luang University. The funders had no role in the study design, data collection and analysis, the decision to publish, or the preparation of the manuscript.

**Competing interests:** The authors have declared that no competing interests exist.

significant gastroprotective benefits and excellent functional and sensory qualities, supporting its potential application in gastrointestinal health–promoting snack formulations.

## 1. Introduction

Peptic ulcer disease (PUD) is a chronic gastrointestinal disease characterized by ulceration of the mucosa extending into the submucosa, occurring most often in the stomach and duodenum. Although its age-adjusted prevalence decreased significantly, from 1990 to 99.4 per 100,000 people in 2019, it remains a substantial health burden, especially in low- and middle-income regions, where mortality and disability-adjusted life years are high [1–4]. These disparities continue to emphasize the need for preventive and supportive management measures aimed at protecting the gastric mucosa.

There has recently been a shift in focus toward functional foods, which have the potential to enhance gastrointestinal health by regulating the inflammatory pathway, mucosal defense, and microbiota balance. Other compounds with potential to help preserve gut integrity and immune and metabolic responses include resistant starches, dietary fiber, and polyphenols [5–7]. Among them, the unripe Musa species used to make green banana flour (GBF) can be singled out as a highly nutritious source of resistant starch, phenolic antioxidants, and non-digestible fiber [8, 9]. Its gastroprotective properties have been demonstrated in several preclinical studies, which show reductions in gastric and intestinal inflammation across various models [10–15]. Moreover, its long-range ethnopharmacological history of use in traditional Thai and Sudanese medicine for the treatment of peptic ulcer disease has long been regarded as evidence of its therapeutic potential [13,16].

Nonetheless, there are challenges associated with the direct application of native GBF in food formulations. Its low solubility, roughness, and low digestibility limit its sensory properties and industrial application [17,18]. Pregelatinization, in particular, is one method of physical modification that provides an effective solution. The drum-drying process, specifically, partially gelatinizes starch granules, thereby improving water solubility, swelling capacity, and handling properties of the flour without any chemical modification of the flour [19–22]. Physicochemical advancements cannot only support product formulation but also affect the bioavailability and potential bioactivity of the final food product.

Although these advantages are well known, a research gap remains regarding the effects of pregelatinization on the gastroprotective activity of GBF and its functional properties in consumer-acceptable edible foods. The study was thus aimed at determining the physicochemical properties of pregelatinized drum-dried GBF, comparing its gastroprotective effects with those of native GBF using an ethanol-induced gastric ulcer rat model, and evaluating shelf-stable snack bars containing pregelatinized GBF in terms of their sensory and storage properties.

We posit that pregelatinization does not alter the gastroprotective performance of GBF but improves its functional and physicochemical characteristics, thereby developing a palatable, stable, health-promoting snack bar that can be used in gastrointestinal nutritional support. These findings may support the use of pregelatinized GBF as a novel multifunctional ingredient in the development of health-promoting food products to improve gastrointestinal health.

## 2. Materials and methods

### 2.1. Preparation of pre-gelatinized green banana flour

Green banana flour (GBF) was prepared from unripe bananas (*Musa sapientum* Linn., ABB group, cv. Kluai Namwa Luang) sourced locally in Thailand and purchased as flour from Musarium Co., Ltd. The preparation protocol adhered to the factory's established guidelines. In summary, fresh bananas were peeled, sliced, and dried at 50°C until their moisture content was reduced to below 10%. Subsequently, they were ground into a fine powder and stored in airtight containers maintained at 4°C until required for use [23].

Pregelatinized GBF was produced utilizing a pilot-scale drum dryer (Model DDM200, Seon Equipment Co., Thailand). The study employed a completely randomized design (CRD) to investigate the effects of GBF concentrations (20%, 30%, 40%, and 50% w/v) on the physicochemical properties of drum-dried formulations designated PregelF1, PregelF2, PregelF3, and PregelF4, respectively. The detailed formulations are presented in Table 1. The mixtures were prepared in accordance with the formulations detailed in Table 2, which included GBF, distilled water, granulated sugar, and table salt. The ingredients were homogenized with a planetary mixer (KitchenAid Professional 5 Plus, USA) at medium speed for 5 minutes until a uniform slurry was achieved. This slurry was subsequently processed using a pilot-scale drum dryer (Model DDM200, Seon Equipment Co., Thailand) with the following operational parameters: Drum gap of $2 \pm 0.5$ mm, blade-to-drum distance of approximately 1 mm, drying temperature of $120 \pm 1$ °C, and drum rotation speed of 0.54 rpm. The dried sheets were scraped off, milled, and sieved through an 80-mesh sieve to obtain a homogeneous powder categorized as pregelatinized GBF (PregelF1, PregelF2, PregelF3, and PregelF4), and stored in sealed polyethylene bags at room temperature until further analyses. Production yield was calculated as the ratio of the dried product weight to the initial slurry weight, expressed as a percentage.

**Table 1. Physicochemical properties of pregelatinized drum-dried green banana flour samples prepared at varying green banana flour concentrations.**

| Parameters* | Starch types | | | | |
|---|---|---|---|---|---|
| | GBF | PregelF1 | PregelF2 | PregelF3 | PregelF4 |
| Concentrations of green banana flour (GBF) (%; w/v) | NA | 20 | 30 | 40 | 50 |
| Yield (%) | NA | 88.900 | 88.000 | 87.100 | 86.900 |
| Water activity | $0.536 \pm 0.001^e$ | $0.293 \pm 0.001^d$ | $0.261 \pm 0.001^c$ | $0.211 \pm 0.001^a$ | $0.219 \pm 0.001^b$ |
| Moisture content (%) | $11.710 \pm 0.200^d$ | $5.960 \pm 0.050^c$ | $5.660 \pm 0.040^b$ | $4.800 \pm 0.070^a$ | $4.830 \pm 0.080^a$ |
| Water solubility index (%) | $5.530 \pm 0.160^a$ | $9.130 \pm 0.950^b$ | $8.230 \pm 0.540^b$ | $7.980 \pm 0.140^b$ | $8.470 \pm 0.340^b$ |
| Swelling power (g/g) | $2.770 \pm 0.120^a$ | $9.930 \pm 0.570^c$ | $8.820 \pm 0.090^b$ | $8.870 \pm 0.030^b$ | $9.120 \pm .0.180^b$ |
| Bulk density (g/mL) | $0.470 \pm 0.010^c$ | $0.350 \pm 0.000^a$ | $0.440 \pm 0.000^b$ | $0.450 \pm 0.000^b$ | $0.460 \pm 0.000^b$ |
| Tap density (g/mL) | $0.550 \pm 0.000^c$ | $0.410 \pm 0.000^a$ | $0.530 \pm 0.000^b$ | $0.530 \pm 0.000^b$ | $0.550 \pm 0.010^c$ |
| Color parameters | | | | | |
| Lightness (*L*\*) | $79.970 \pm 0.190^c$ | $75.100 \pm 0.550^{ab}$ | $74.820 \pm 0.650^{ab}$ | $74.680 \pm 0.540^a$ | $75.310 \pm 0.390^b$ |
| Redness (*a*\*) | $2.130 \pm 0.020^a$ | $3.410 \pm 0.140^b$ | $3.490 \pm 0.130^b$ | $3.470 \pm 0.150^b$ | $3.490 \pm 0.070^b$ |
| Yellowness (*b*\*) | $10.560 \pm 0.120^a$ | $11.950 \pm 0.160^c$ | $11.940 \pm 0.190^c$ | $11.960 \pm 0.270^c$ | $11.670 \pm 0.140^c$ |

*Values are expressed as mean±SD (n = 3). Different superscripts within rows indicate significant differences (p < 0.05) across starch types. GBF, green banana flour; NA, not applicable.

**Table 2. Physicochemical characteristics and sensory evaluation scores of snack bars formulated with varying proportions of pregelatinized drum-dried green banana flour.**

| Parameters | Samples of the snack bars | | | |
|---|---|---|---|---|
| | sPreF1 | sPreF2 | sPreF3 | sPreF4 |
| Water activity | 0.376±0.003$^c$ | 0.332±0.001$^b$ | 0.327±0.000$^a$ | 0.335±0.001$^b$ |
| Moisture content (%) | 5.390±0.160$^c$ | 4.413±0.070$^b$ | 3.630±0.030$^a$ | 3.797±0.050$^a$ |
| Color parameters | | | | |
| Lightness (L*) | 59.170±1.680$^b$ | 60.810±2.470$^c$ | 60.300±1.870$^{bc}$ | 53.680±2.200$^a$ |
| Redness (a*) | 3.970±0.450$^a$ | 4.080±0.320$^a$ | 4.110±0.380$^a$ | 4.780±0.430$^b$ |
| Yellowness (b*) | 11.960±1.200$^{ab}$ | 12.460±0.790$^b$ | 12.070±0.770$^b$ | 11.410±0.890$^a$ |
| Sensory aspects (n=30) | | | | |
| Color | 6.270±1.560$^a$ | 6.300±1.510$^a$ | 6.900±1.440$^a$ | 6.470±1.710$^a$ |
| Crunchiness | 7.170±1.270$^{ab}$ | 7.600±1.220$^b$ | 7.470±0.860$^{ab}$ | 6.870±1.410$^a$ |
| Flavor | 6.600±1.240$^a$ | 6.600±1.480$^a$ | 6.930±1.180$^a$ | 6.630±1.350$^a$ |
| Taste | 6.870±0.920$^a$ | 7.030±1.050$^a$ | 7.300±0.870$^a$ | 6.870±1.260$^a$ |
| Overall appearance | 6.800±1.080$^a$ | 7.130±1.150$^a$ | 7.400±0.930$^b$ | 6.800±1.300$^a$ |

*Values represent mean±SD (n=3, sensory evaluation n=30). Different superscripts within rows indicate significant differences (p<0.05) across snack formulations.

## 2.2. Physicochemical analyses of pregelatinized GBF

The physicochemical properties of pregelatinized GBF include moisture content, water activity, bulk density, water solubility index (WSI), swelling capacity, and color analysis [24,25].

**2.2.1. Moisture content.** Moisture content was determined by oven-drying samples at 105 °C for 4 h until constant weight [25].

**2.2.2. Water activity.** Water activity was measured at 25 °C using an AquaLab Series 4TE meter (Decagon Devices, Inc., USA), with powdered samples equilibrated in disposable cups [25].

**2.2.3. Bulk and tapped density.** Bulk and tapped densities were determined following standard gravimetric procedures as described by Nakorn et al. [24]. Briefly, a 3g sample was weighed into a 10 mL graduated cylinder, and the initial volume was recorded to calculate bulk density (g/mL) as the ratio of sample weight to volume. The cylinder was then tapped ten times to achieve volume compaction, and the tapped density was calculated as the ratio of sample weight to the reduced volume (g/mL).

**2.2.4. Water solubility index (WSI).** Their WSI was determined by mixing 1 g of the sample in 10 mL of distilled water, stirring for 30 minutes, centrifuging at 3,000 rpm for 10 minutes, drying the supernatant at 105°C, and calculating the percentage [25].

**2.2.5. Swelling power.** For swelling power, 1 g of the sample was added to 10 mL of distilled water in a graduated cylinder and left undisturbed for 24 hours. The final volume was recorded, and swelling power was calculated as the ratio of the swollen volume to the initial sample weight (g/g) [25].

**2.2.6. Color measurement.** Color parameters (L*, a*, b*) were determined using a ColorFlex EZ colorimeter (HunterLab, USA) calibrated with a white standard tile. Triplicate readings were taken from different areas of each sample to obtain average color values [25].

## 2.3. Formulation of pregelatinized green banana flour snack bars and their storage stability

The banana flakes from the four formulations were formed into snack bars using a binding solution of 83% syrup and 17% clean water. This binding solution was added at 40% (w/w) relative to the total weight of the banana flakes (100%). The

                                                      

mixture was thoroughly blended with a mechanical mixer for 5 minutes. It was then pressed into pre-prepared rectangular molds (3 × 9 cm$^2$), ensuring each bar weighed precisely 15 grams. Finally, the bars were baked at 170°C for 25–30 minutes. After cooling, the bars were packed individually in opaque aluminum foil bags and stored at 25°C with 50–70% relative humidity (RH). Following baking, the snack bars were subjected to physicochemical analyses, including moisture content, water activity, and color evaluation, as mentioned earlier.

## 2.4. Sensory evaluation

Additionally, a sensory evaluation was performed utilizing a 9-point hedonic scale, involving 30 trained panelists and focusing on five parameters: color, crunchiness, flavor, taste, and overall appearance. The 9-point hedonic scale was anchored as follows: 1 = dislike extremely, 2 = dislike very much, 3 = dislike moderately, 4 = dislike slightly, 5 = neither like nor dislike, 6 = like slightly, 7 = like moderately, 8 = like very much, and 9 = like extremely.

The attributes evaluated were color, crunchiness, flavor, taste, and overall appearance. Samples from each formulation (sPreF1–sPreF4) were presented in randomized order under uniform lighting conditions. Panelists were provided with water to rinse between samples. Evaluations were performed immediately after production and at designated intervals during storage (0, 2, 4, and 6 weeks) to assess changes in sensory quality.

## 2.5. Microbiological quality

Microbiological quality was evaluated by total viable counts (TVC), yeast, and mold counts. Samples were serially diluted in sterile saline and plated on growth media (Plate Count Agar for TVC, Potato Dextrose Agar for yeast and molds). Plates were incubated at 37°C for 48 hours for TVC and at 25°C for 5–7 days for yeast and mold enumeration. Results were reported as colony-forming units per gram (CFU/g) [26,27]. Lipid oxidation was monitored by measuring peroxide values via standard iodometric titration, expressed as milliequivalents of peroxide-oxygen per kilogram (meq/kg) [28].

## 2.6. The gastroprotective effect of GBF and PregelF3 against ethanol-induced acute gastric lesions in rats

Male Wistar rats, weighing 180–220 grams, were sourced from the National Laboratory Animal Center at Mahidol University in Thailand. The rats underwent a 7-day acclimatization period in standard laboratory conditions (22±2°C, 12-hour light/dark cycle). They were provided with access to a standard nutritionally balanced commercial diet (No. C.P. 082, Perfect Companion Group Co. Ltd., Thailand) and water ad libitum. All experimental protocols involving animals were approved by the Institutional Animal Care and Use Committee (IACUC), Prince of Songkla University (Protocol No. 2562-01-009; MOE 0521.11/563; Ref. 41/2019).

Humane endpoints were established before the study began. Animals were checked at least twice daily for signs of pain, distress, infection, or other adverse reactions related to treatment or wounding. Humane euthanasia criteria included persistent weight loss (>20%), ulceration or necrosis at the wound site, decreased mobility, excessive grooming or vocalization, abnormal posture, and lack of response to external stimuli. Animals showing these signs were euthanized immediately with an overdose of pentobarbital sodium (≥150 mg/kg, intraperitoneally), following AVMA guidelines. No unexpected deaths occurred during the study, and all animals reached their predetermined endpoints.

Rats were randomly allocated into five groups (n = 6/group):

(1) sham control (received distilled water instead of ethanol; this group did not undergo ulcer induction);

(2) vehicle control (received distilled water for seven days, followed by ethanol to induce ulcers; this group serves as the ulcerated negative control to compare treatment effects);

(3) positive control (omeprazole (20 mg/kg) is the standard anti-ulcer drug used in ethanol-induced ulcer models to validate the model and benchmark the protective efficacy of test substances);

(4)  pregelatinized GBF (PregelF3, 1000 mg/kg); and

(5)  native GBF (1000 mg/kg).

The test substances were administered orally once daily for a duration of seven consecutive days. Both GBF and PregelF3 powders were freshly suspended in distilled water immediately before dosing to obtain a homogeneous slurry suitable for oral administration. The suspensions were administered to the rats by oral gavage at a dose of 10 mL/kg body weight to ensure consistent delivery of the 1000 mg/kg dose. This method ensured accurate dosing, minimized the risk of aspiration, and complied with standard procedures for safe oral administration in rodents. On the sixth day, the animals were subjected to an overnight fasting period, and on the seventh day, one hour following the final administration, gastric ulcers were induced by means of oral gavage using 80% ethanol (1 mL/rat), with the exception of the sham group, which received distilled water instead. Animals were euthanized four hours after ulcer induction. After the stomachs were excised and opened along the greater curvature, the gastric contents were gently collected by rinsing the stomach lumen with a fixed volume of distilled water (2 mL) using a sterile Pasteur pipette into pre-labelled tubes. The pH of each sample was immediately measured using a calibrated pH meter (Thermo Fisher Scientific, MA, USA). The gastric mucosa was photographed and analyzed using ImageJ to quantify ulcer area, from which the ulcer index (UI) and percentage of ulcer protection were calculated [29].

The content of gastric mucin was determined through an Alcian blue dye-binding assay. Gastric mucosal samples were homogenized and treated with Alcian blue solution, and the absorbance was recorded at 598 nm. The results are presented as µg of Alcian blue equivalents per gram of gastric tissue. Gastric tissues were fixed in 10% buffered formalin, embedded in paraffin, sectioned to a thickness of 5 µm, and stained with Hematoxylin and Eosin (H&E). A microscopic examination was conducted to evaluate epithelial integrity, inflammatory infiltration, and the overall mucosal architecture.

## 2.7. Statistical analysis

Data were expressed as mean ± standard deviation (SD) or standard error of the mean (SEM), as appropriate. Statistical analyses were performed using one-way analysis of variance (ANOVA) followed by Duncan's multiple range test to determine differences among group means using SPSS software (version 22.0; SPSS Inc., Chicago, IL, USA). Pairwise comparisons were made between each treatment group (native GBF and PregelF3) and both the vehicle control and positive drug control (omeprazole, 20 mg/kg) to determine the relative efficacy of each intervention. Differences were considered significant at $p < 0.05$. Groups sharing the same superscript letter (in tables) or without distinct symbols (in figures) were not significantly different.

## 3.  Results

### 3.1.  Physicochemical properties of pregelatinized green banana flour

As shown in Table 1, the physicochemical properties of GBF were compared among four pregelatinized formulations (PregelF1–PregelF4) containing increasing concentrations (20%–50% w/v) of GBF. These results clearly indicate that pre-gelatinization significantly differed among the groups (Statistical differences among the groups were assessed using ANOVA followed by Duncan's post hoc test). Values with different superscript letters in the same row or column indicate significant differences ($p < 0.05$), which altered the physicochemical properties of GBF, including water activity, moisture content, water solubility index, swelling power, bulk and tap density, and color attributes. Increasing GBF concentrations (20–50% w/v) resulted in a gradual reduction in yield from 88.9% (PregelF1) to 86.9% (PregelF4). The decline in yield at higher concentrations is likely due to increased viscosity and stickiness of the GBF slurry, which promotes greater adhesion to the drum surface during drying and thereby reduces product recovery. This indicates that higher GBF concentrations could reduce processing efficiency, likely due to increased viscosity or adhesion during drum drying. Moreover, the pre-gelatinized treatment significantly lowered the water activity and moisture content of GBF flake, potentially improving

microbial safety and shelf stability. An approximately 2-fold reduction in water activity (from 0.536 to 0.211–0.293) and moisture content (from 11.71% to 4.80–5.96%) was observed in pre-gelatinized GBF compared to its original form. The lowest values recorded for water activity (0.211) and moisture content (4.80%) in PregelF3 indicate optimal dehydration at 40% GBF. Additionally, the pre-gelatinization process significantly improved water solubility (5.53% in GBF to approximately 8–9% in pregelatinized samples) and water absorption capacity (from 2.77 g/g in GBF to about 9 g/g in pregelatinized samples), suggesting that this process could effectively enhance solubility and improve the water-holding capacity of GBF.

Conversely, the bulk and tap densities show only slight changes, indicating that the drum-dried pre-gelatinization method is unlikely to affect porosity or improve packing capacity. As expected, the color analysis showed slight reductions in lightness ($L^*$) and increased redness ($a^*$) and yellowness ($b^*$) in pregelatinized samples, indicating Maillard reactions or pigment changes during thermal processing. In summary, drum drying pre-gelatinizes GBF, enhancing its attributes such as water solubility, swelling capacity, and moisture stability, making it suitable for functional foods. Data suggest PregelF3 (40% GBF) offers an optimal balance of functional properties and processing efficiency, with minimal color degradation and low water activity.

## 3.2. Physicochemical properties and sensory evaluation of snack bars containing pregelatinized green banana flour

To explore the potential of its application as a functional food, a snack bar made from four types of pregelatinized green banana flour was prepared and evaluated for its physicochemical properties and sensory attributes, as detailed in Table 2. As critical indicators of shelf stability and microbial safety, the water activity and moisture content of the formulated snack bars were found to be approximately 0.3 and 3.6–5.4%, respectively. The lowest water activity and moisture content, at 0.327 and 3.63%, respectively, were observed in the formulated snack bar made with PregelF3 Flour (sPreF3). Color parameters influenced by thermal processing were further examined (Table 2 and Fig 1). As expected, sPreF4, made from the pre-gelatinized flour with the highest GBF content, exhibited the lowest lightness ($L^*$) values of 53.68 and the highest redness ($a^*$) of 4.78, indicating significant Maillard browning at elevated GBF levels. Based on the sensory attributes obtained from the trained panelists, there are no significant differences among the formulations in terms of color, flavor, taste, and overall appearance, except for crunchiness. Among the formulations, sPreF3 received numerically higher ratings for color, flavor, taste, and overall appearance; however, these differences were not statistically significant ($p > 0.05$). It was, nonetheless, significantly preferred for crunchiness compared to sPreF4 ($p < 0.05$). Overall, while sensory ratings were comparable across formulations, sPreF3 tended to score higher in most sensory attributes and was significantly preferred for crunchiness, suggesting a more desirable texture at this formulation level.

## 3.3. Gastroprotective effects of GBF and PregelF3 in ethanol-induced gastric ulcer model

To further explore the impact of the drum-dried pre-gelatinization method on the gastroprotective qualities of native GBF, Table 3 outlines the effects of both native GBF and its pregelatinized drum-dried variation (PregelF3) on ethanol-induced gastric mucosal lesions in rats. No significant differences were observed in the total gastric area across the ethanol-treated groups (vehicle, omeprazole, PregelF3), reflecting similar anatomical variations. The vehicle-treated rats had the largest total ulcer area ($0.41 \pm 0.09$ cm²), confirming the ulcerogenic effect of absolute ethanol. Administration of omeprazole (20 mg/kg), the positive control, significantly reduced the ulcer area to $0.04 \pm 0.01$ cm² ($p < 0.05$), confirming the efficacy of the positive control and the model's sensitivity. Both GBF and PregelF3 showed significant protective effects, decreasing ulcer area to $0.02 \pm 0.01$ cm² and $0.09 \pm 0.03$ cm², respectively ($p < 0.05$ vs. vehicle). Macroscopic images clearly illustrate the gastroprotective effects of GBF and PregelF3 against ethanol-induced mucosal damage. Normal control rats exhibited intact gastric mucosa, whereas significant mucosal lesions were evident in gastritis controls. GBF and PregelF3 treatment effectively reduced mucosal damage, closely resembling the positive control group treated with

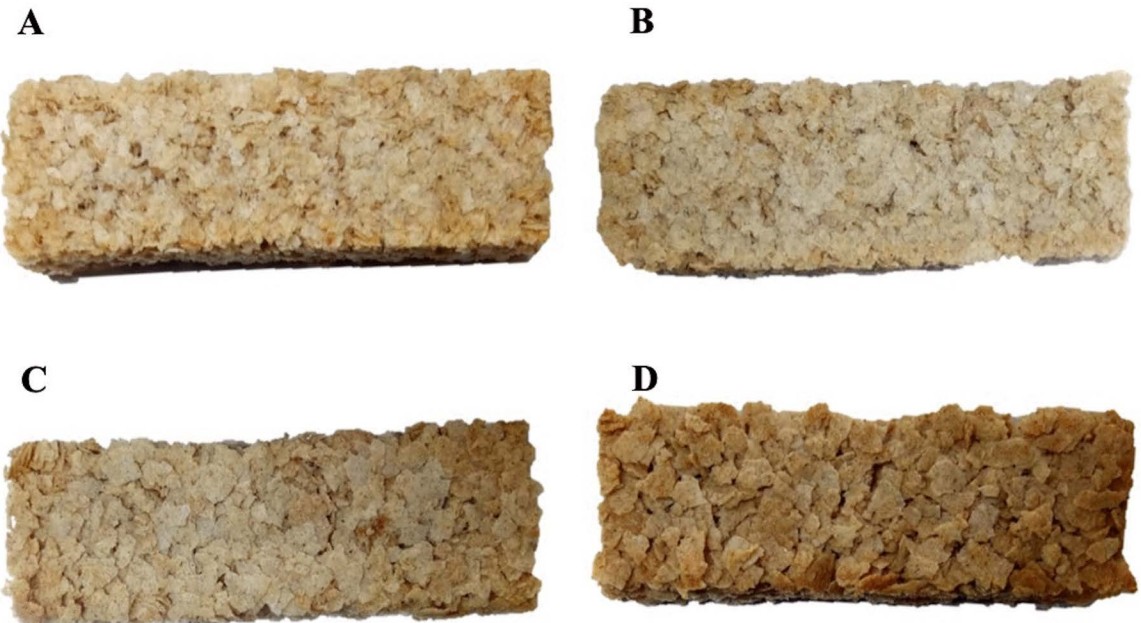

**Fig 1. The visual appearance of snack bars formulated with pregelatinized drum-dried green banana flour at concentrations of 20% (sPreF1; A), 30% (sPreF2; B), 40% (sPreF3; C), and 50% (sPreF4; D) w/w.**

**Table 3. Effects of green banana flour and pregelatinized drum-dried green banana flour (PregelF3) on ethanol-induced gastric mucosal lesions in rats.**

| Experimental groups | Total gastric area (cm²) | Total ulcer area (cm²) | Ulcer index |
|---|---|---|---|
| Sham-feeding (control) | 11.600 ± 1.250 | ND | NA |
| Vehicle control | 16.570 ± 0.730 | 0.410 ± 0.090# | 0.830 ± 0.060# |
| Omeprazole (20 mg/kg) | 17.880 ± 0.710 | 0.040 ± 0.010* | 0.600 ± 0.080* |
| PregelF3 (1,000 mg/kg) | 16.390 ± 0.670 | 0.090 ± 0.030* | 0.630 ± 0.050* |
| green banana flour (1,000 mg/kg) | 15.790 ± 0.630 | 0.020 ± 0.010* | 0.750 ± 0.110* |

Values are expressed as mean ± SEM (n = 6). * and # indicate significant differences from the vehicle control and sham-feeding groups, respectively (* vs. vehicle; # vs. sham) $p < 0.05$; one-way ANOVA followed by Duncan's multiple range test.

omeprazole (Fig 2). As shown in Fig 3A, rats in the vehicle control group exhibited a significantly higher ulcerated area (approximately 2.8%) compared to the sham-fed group, which had no detectable ulcers. Pre-treatment with either GBF or PregelF3 significantly reduced the ulcerated area to approximately 0.2–0.5%, demonstrating comparable protective effects to omeprazole (0.1%). A significant decrease in gastric pH (~pH 2.8) and mucin content (~20 µg/g tissue) was observed in ethanol-treated vehicle controls, indicating enhanced acid secretion and mucosal vulnerability (Fig 3B and 3C). The consumption of both GBF and PregelF3 significantly raised gastric pH to ~4.5–5.0, demonstrating their ability to either buffer gastric acid or reduce acid secretion. Additionally, these flours may enhance mucus production or preserve mucin, as both GBF and PregelF3 significantly restored mucin levels to approximately 26–27 µg/g. While treatment with omeprazole restored pH and mucin levels to approximately 6.0 and 24 µg/g, respectively. Histological examination (Fig 4) confirmed the gastroprotective effects of GBF and PregelF3. Ethanol-induced gastric damage in vehicle controls showed severe inflammatory cell infiltration, epithelial erosion, and mucosal compromise. Treatment with GBF and

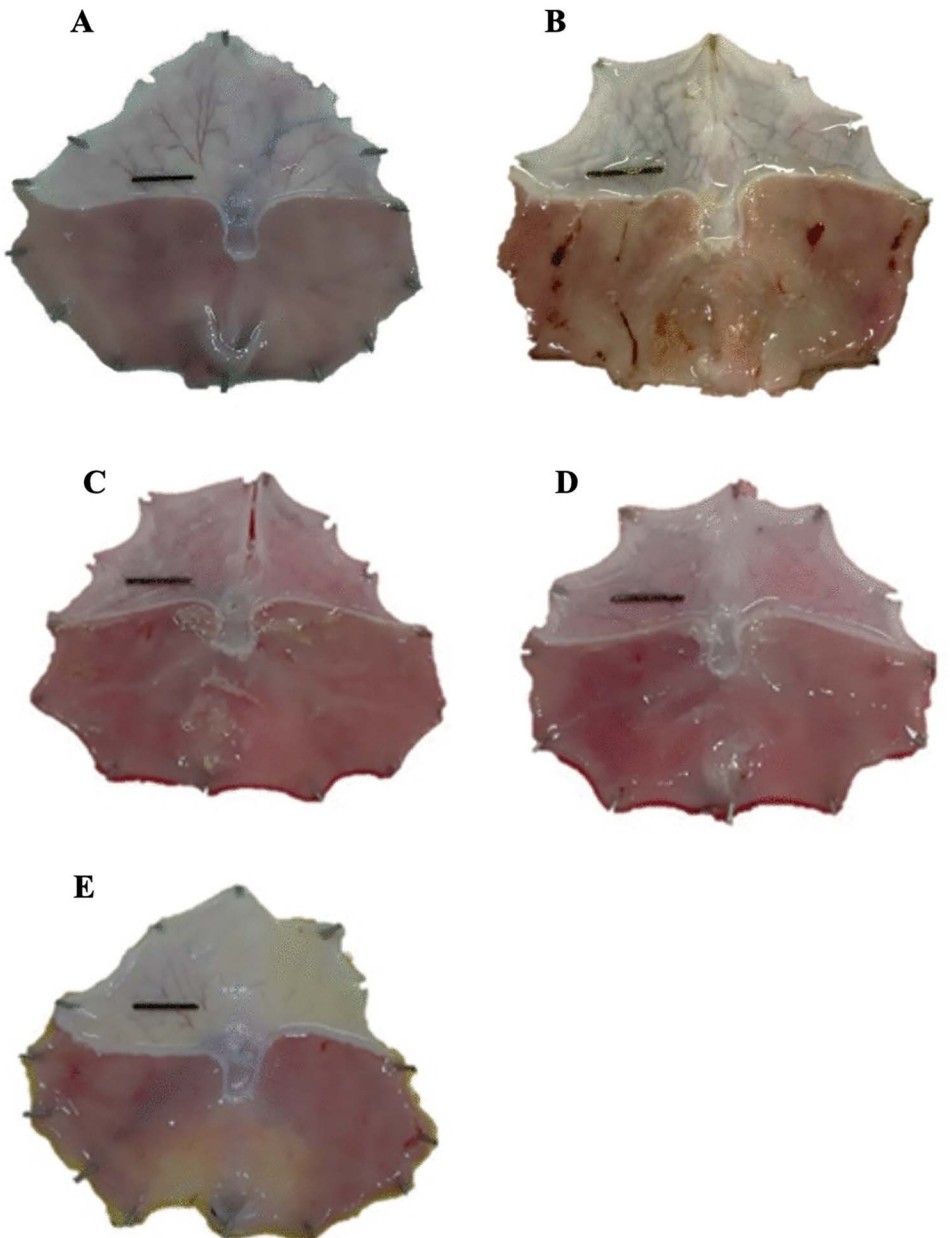

**Fig 2. Macroscopic appearance of gastric mucosal lesions induced by ethanol in rats after treatment with green banana flour (GBF, 1000 mg/kg) and pregelatinized drum-dried green banana flour (PregelF3, 1000 mg/kg).** Representative images from (A) standard control, (B) gastritis control, **(C)** GBF-treated, **(D)** PregelF3-treated, and (E) omeprazole-treated groups (20 mg/kg) are shown.

PregelF3 reduced inflammation, preserved tissue architecture, and improved mucosal integrity, similar to omeprazole-treated rats.

Therefore, it is noteworthy that the administration of **both** GBF and PregelF3 for seven consecutive days at a dosage of 1,000 mg/kg resulted in a substantial reduction in the total ulcer area and ulcer index, with efficacy comparable to that of the standard anti-ulcer medication, omeprazole. These effects potentially occur through the neutralization of gastric acid

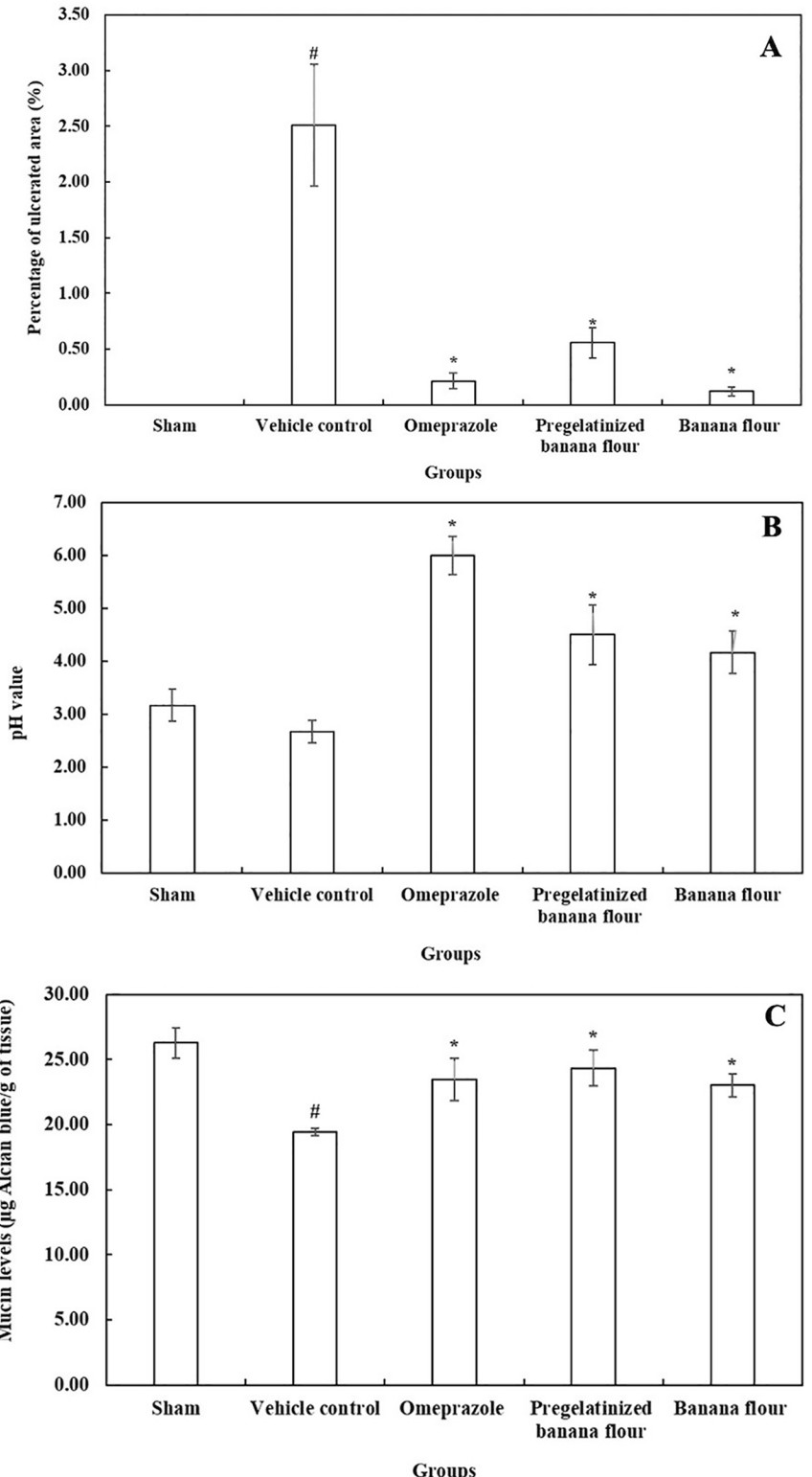

**Fig 3. Quantitative gastroprotective effects of green banana flour and pregelatinized drum-dried green banana flour (PregelF3) (A) gastric ulceration area, (B) gastric pH levels, and (C) mucin content in ethanol-induced gastric ulcers in rats.** Values are expressed as mean ± SEM (n = 6). * and # indicate significant differences from the vehicle control and sham-feeding groups, respectively (* vs. vehicle; # vs. sham) $p < 0.05$; one-way ANOVA followed by Duncan's multiple range test.

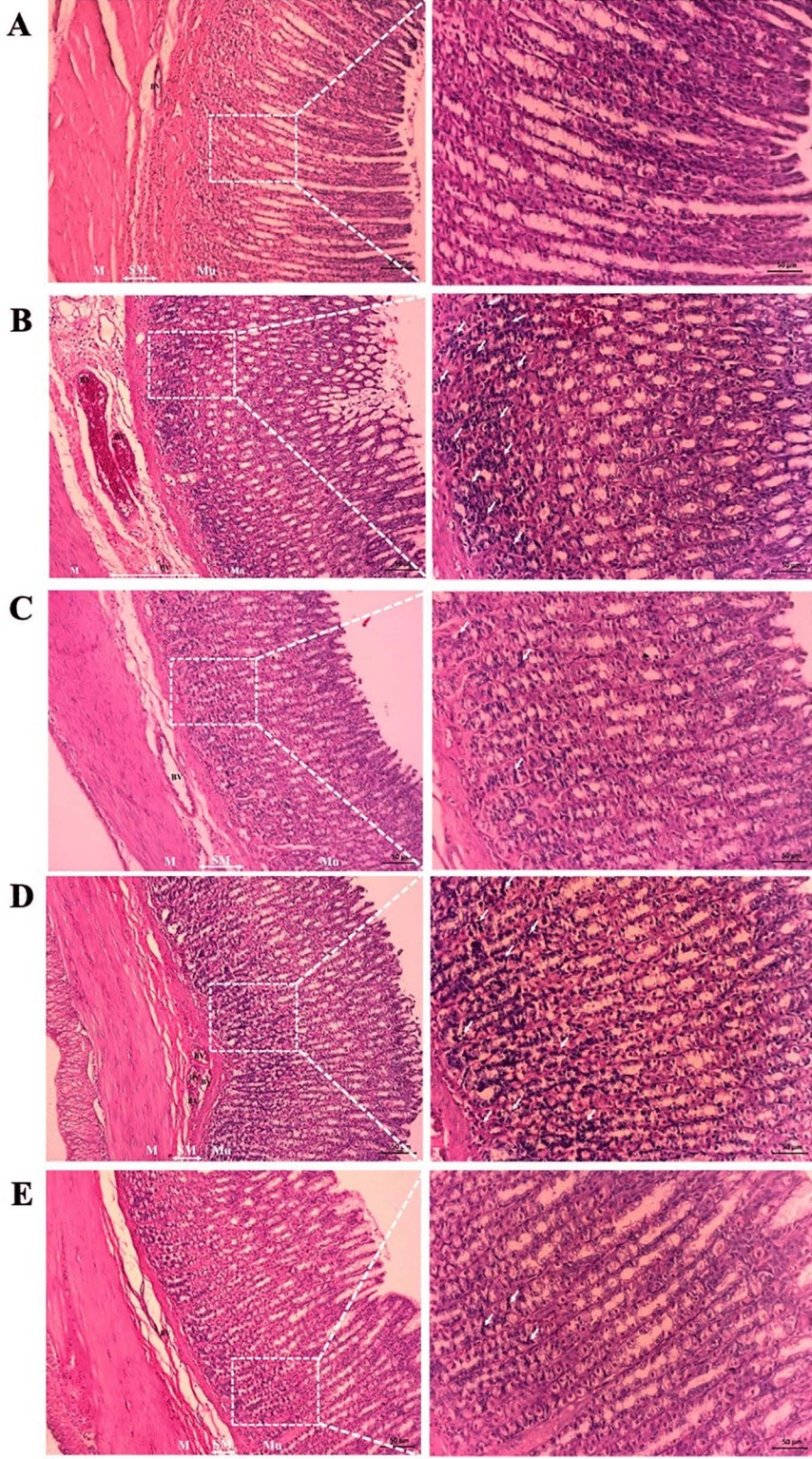

**Fig 4. Representative photomicrographs of sections from rat gastric mucosa in ethanol-induced gastric ulcers (magnification at 10× in the left panel and 20× in the right panel).** Green banana flour (1,000 mg/kg/day p.o.: **E**), pregelatinized drum-dried green banana flour (PregelF3) (1,000 mg/kg/day p.o.: **D**), and omeprazole (20 mg/kg/day p.o.: C) were orally administered starting one week prior to ulcer induction with an 80% ethanol solution (1 mL/animal). Sham-feeding animals (A) and vehicle **(B)**-treated animals served as controls. White arrows indicate the inflammatory cells present in the lamina propria. Mu: mucosa layer, SM: submucosa layer, M: muscle layer, BV: blood vessel.

and the preservation of mucosal defenses. The comparable efficacy observed between native and pregelatinized flour suggests that the pregelatinization process does not disrupt the functional properties of GBF.

### 3.4. Shelf stability, microbiological quality, and sensory characteristics of pregelatinized green banana flour-based snack bars

The shelf-stability parameters of sPreF3 snack bars during a six-week storage period at ambient conditions (25°C, 50–70% RH) are presented in Table 4. The results indicated that moisture uptake occurred during storage, as evidenced by a gradual increase in water activity from 0.276 to 0.291 and a corresponding rise in moisture content from 3.19% to 4.15%. An acceptable microbiological quality was also achieved in the formulated snack bar, with total viable microbial counts stabilizing around $10^2$ CFU/g after the second week. In comparison, yeast and mold counts remained generally low (<10 CFU/g). At week 0, total viable counts were below the detection limit (<10 CFU/g). The slight increase to approximately $10^2$ CFU/g observed by week 2 may be attributed to minor post-baking contamination during handling or packaging, which is common in low-moisture bakery-type products. However, the microbial counts remained stable and well within acceptable limits for ready-to-eat snack products throughout the six-week storage period, indicating adequate hygienic handling and effective packaging integrity.

Despite this increase, microbial levels remained very low throughout the six-week storage period, consistent with acceptable limits for baked, low-moisture products. The low water activity (0.27–0.29) and moisture content (3–4%) of the snack bars effectively restricted microbial proliferation, maintaining the product's microbiological safety during storage. The peroxide values remained consistently low from week two onward, ranging from 0.30 to 0.32, indicating stable lipid quality with negligible oxidative degradation.

Sensory evaluations consistently showed high acceptance scores, with minor fluctuations in color (moderately), crunchiness (from moderately to very much), flavor (from slightly to moderately), taste (from moderately to very much), and overall appearance (from moderately to very much). This suggests that storage may slightly influence consumer perception. Sensory evaluation scores showed statistically significant but minor fluctuations over the six-week period, particularly in crunchiness, flavor, and overall appearance. These changes may be related to gradual moisture absorption and minor

**Table 4. Shelf stability, microbiological quality, and sensory characteristics of pregelatinized green banana flour-based snack bars (sPreF3) stored at 25°C and 50-70% relative humidity over six weeks.**

| Parameters | Storage time (at 25°C; 50–70% RH in opaque aluminium foil bags*) | | | |
|---|---|---|---|---|
| | Initial | 2nd week | 4th week | 6th week |
| Water activity | 0.276 ± 0.053 | 0.275 ± 0.034 | 0.289 ± 0.017 | 0.291 ± 0.038 |
| Moisture content (%) | 3.191 ± 0.060 | 3.975 ± 0.010 | 4.105 ± 0.050 | 4.155 ± 0.040 |
| Total viable count (CFU/g) | <10 | $1.1 \times 10^2$ | $1.2 \times 10^2$ | $1.1 \times 10^2$ |
| Yeast and mold count (CFU/g) | <10 | <10 | <10 | <10 |
| Peroxide value (the amount of peroxide-oxygen per 1 kg) | ND | 0.302 | 0.315 | 0.315 |
| Sensory aspects (*n* = 30; mean± SD) | | | | |
| Color | 7.000 ± 1.390[a] | 7.200 ± 1.080[a] | 7.750 ± 0.830[b] | 7.750 ± 0.890[b] |
| Crunchiness | 7.800 ± 0.830[b] | 7.150 ± 1.310[c] | 8.150 ± 0.570[a] | 7.500 ± 1.070[b] |
| Flavor | 7.230 ± 1.410[b] | 6.250 ± 1.610[a] | 7.450 ± 1.120[b] | 6.850 ± 1.530[a] |
| Taste | 7.500 ± 1.150[b] | 7.050 ± 1.070[a] | 8.000 ± 0.770[c] | 7.550 ± 1.070[b] |
| Overall appearance | 7.570 ± 0.880[b] | 7.200 ± 1.120[a] | 8.050 ± 0.860[c] | 7.450 ± 1.280[b] |

* The ASTM E96: a water vapor permeability rate of the opaque aluminium foil bags at 25°C; 50−70% RH was $6.61 \times 10^{-4}$ g/m²/day.cm².mmHg. CFU/g, colony-forming units per gram; ND, not detected. Different superscripts within rows indicate significant differences (p < 0.05) across storage times.

shifts in texture or flavor perception over time. Nevertheless, the overall sensory acceptability remained high, confirming the product's stability and consumer appeal during storage.

## 4. Discussion

This study examines the use of pregelatinized green banana flour (GBF), derived from drum drying, as a functional ingredient in the formulation of snack bars, with a focus on its physicochemical properties, sensory acceptability, gastroprotective effects, and storage stability. The drum-drying pre-gelatinization process significantly improved the physicochemical properties of the GBF, specifically by reducing moisture content and water activity, which are crucial for microbial safety and product shelf-life.

The process of heating green banana flour via drum drying improves its water solubility and swelling capacity, as evidenced by this study, due to various physicochemical transformations that take place during this procedure. Thermal treatment can increase the hydrolysis of starch molecules, further contributing to the solubility of the flour [22,30,31]. Additionally, the enhancement of the solubility and swelling properties of GBF may result from the partial gelatinization of starch induced by heat treatment. This enhancement may be attributed to the disruption of starch granule crystallinity and the partial gelatinization occurring during drum drying, which exposes amorphous regions and facilitates greater water penetration and molecular mobility. Consequently, the flour exhibits improved hydration behavior and solubility, which are desirable properties in product formulations requiring enhanced texture and viscosity control. Such physicochemical transformations corroborate the reported thermally induced structural modifications in other starch-based systems subjected to similar thermal treatments [22,30–32]. This finding aligns with previous studies, which demonstrate that increased temperature during drying processes substantially boosts the swelling power and water solubility index of green bananas [33,34]. These modifications make the flour more suitable for applications in food products where high-water absorption and swelling are desirable, such as bakery and confectionery items [35,36]. Overall, the heat treatment of green banana flour through drum drying effectively modifies its structural and functional properties, leading to increased water solubility and swelling power, which are beneficial for various food industry applications.

Different drying methods, including freeze-drying, hot air drying, and Refractance Window Drying (RWD), affect GBF properties in various ways. Freeze-dried banana flour typically has a higher resistant starch content, which benefits nutritional purposes, although it may result in a higher glycemic index in products like gluten-free bread [37,38]. Additionally, RWD produces flour with lower water activity, a higher whiteness index, and improved flowability compared to hot air-drying methods, indicating superior quality in terms of appearance and handling [34]. Drum drying offers significant advantages over the techniques mentioned above. It is highly energy-efficient, especially for drying viscous pastes or purees, making it economically attractive for the food industry [39,40]. Drum-dried products typically exhibit desirable characteris Walailak Unive Walailak Unive Walailak Unive Walailak Unive Walailak Unive Walailak Unive Walailak Unive Walailak Univetics, such as flaky textures and improved rehydration properties, which are ideal for foods like cereals and dairy-based items [41]. Drum drying supports various production scales, yielding faster and cheaper results than freeze-drying. It provides better heat distribution than hot air drying and RWD [42]. While it preserves slightly fewer nutrients and sensory qualities than freeze-drying, its efficiency, versatility, and quality often make it preferable [41–43]. Sensory evaluations indicated that snack bars containing 40% pregelatinized GBF (sPreF3) received numerically higher ratings across most sensory attributes and were significantly preferred for crunchiness compared to Walalak UnivesPreF4. This formulation demonstrated a favorable balance of texture, flavor, and appearance that likely contributed to its higher overall acceptability. Although differences in color, flavor, taste, and overall appearance were not statistically significant, sPreF3 consistently showed a favorable trend in sensory perception. This suggests that incorporating pregelatinized GBF at this concentration contributes to an optimal texture and a balanced sensory profile, likely due to suitable moisture management and desirable Maillard reaction products developed during baking [44,45]. Furthermore, even though this product contains up to 40% of pregelatinized GBF, the sensory attributes and overall acceptability of this product were comparable to noodles

containing approximately 10% of banana starch [46,47]. The microbiological results showed that the total viable counts were less than the detection limit (<10 CFU/g) at week 0, which was recorded as ND in Table 3. The following low counts (~$10^2$ CFU/g) at week 2 likely reflect minor post-baking contamination introduced during handling or packaging and not due to microbial growth. Given the low moisture content (3–4%) and water activity (~0.27–0.29), conditions were highly unfavorable for microbial growth, consistent with the expected stability of baked, low-moisture products. According to established microbial safety criteria for bakery and snack products, total aerobic counts of less than $10^3$ CFU/g are considered microbiologically fit [48]. Walailak Unive

The gastroprotective effects of GBF against peptic ulcers are attributed to a complex biochemical mechanism with various active components. The GBF is rich in polyphenols, flavonoids, and resistant starch, which enhances its protective properties [11–13]. These compounds enhance the defensive properties of the gastric mucosa by promoting mucosal growth and strengthening the mucous-phospholipid layer, which serves as a barrier against ulcerogenic agents such as aspirin and indomethacin [13,49,50]. Ethanol-induced gastric ulcer models were used to investigate whether the pre-gelatinization process influences the gastroprotective effect against gastric ulcers. In this model, ethanol directly harms the gastric mucosa, resulting in ulcer formation, and consistently produces the largest ulcer surface area compared to aspirin and indomethacin [51]. Importantly, this study demonstrates for the first time that the pregelatinization process, specifically drum drying, does not diminish the gastroprotective efficacy of green banana flour. Both the native and pregelatinized forms exhibited comparable reductions in ulcer area, restoration of mucin content, and normalization of gastric pH, underscoring that the structural modifications introduced during pregelatinization preserve the bioactive components responsible for mucosal protection. Although this is the first report examining the gastroprotective effects of pregelatinized GBF, prior research has shown that the components phosphatidylcholine and pectin found in bananas help protect the gastric mucosa by strengthening the mucous layer and possibly regulating prostaglandin levels [49,52]. This aligns with our study, which observed a decrease in the area of gastric ulceration and recorded an increase in mucus content. These levels are crucial for maintaining mucosal integrity and promoting healing [13,49,52]. Ethanol intake damages the gastric mucosa by causing oxidative stress, which leads to the production of reactive oxygen species (ROS) and weakens antioxidant defenses like glutathione (GSH) and catalase [53,54]. The polyphenols and flavonoids found in GBF may offer protective benefits against ethanol-related peptic ulcers. These substances have antioxidant effects that enhance enzymatic antioxidants, such as superoxide dismutase (SOD), and non-enzymatic antioxidants like GSH, while also reducing oxidative stress markers, including malondialdehyde (MDA) [55–57].

Even though our results highlight the gastroprotective effect of GBF and its pregelatinized flour for the first time and demonstrate their utilization in snack bars, some limitations should be mentioned. Shelf-life assessments employ relatively short durations and specific storage conditions, which may not accurately reflect actual storage scenarios. Future investigations should include extended storage durations under various environmental conditions, as well as larger and more diverse sensory panels that were relatively limited in this study. Although the gastroprotective properties of both GBF and pregelatinized GBF were confirmed in this work, comprehensive studies on their mechanisms of action at both molecular and biochemical levels should be conducted to thoroughly clarify this effect.

In summary, it should be highlighted that the pregelatinization process used in this study does not attenuate the gastroprotective properties of GBF. The pregelatinized GBF developed in this present work serves as a useful and effective functional ingredient in snack bar production. It offers enhanced physicochemical properties, increases consumer appeal, and possibly extends the shelf life of the product. Furthermore, the pre-gelatinization process utilizing drum drying does not affect the gastroprotective properties of GBF. Consequently, it presents significant potential for developing functional foods that promote gastrointestinal health. Although biochemical parameters such as cytokines and oxidative stress markers were not assessed in this study, future investigations should include these analyses to delineate the molecular pathways underlying the observed gastroprotective effects.

## 5. Conclusion

This study shows pregelatinized green banana flour (GBF), made via drum drying, as a potential functional food ingredient with gastroprotective properties. The pregelatinization improved GBF's water solubility and swelling, boosting its food application. Drum-drying retained GBF's gastroprotective effects, matching the native form's efficacy in a gastric ulcer model, making it suitable for gastrointestinal health formulations. Snack bars with 40% pregelatinized GBF (sPreF3) had high sensory acceptance, low water activity, and good shelf life at room temperature. These findings indicate that drum-dried GBF is an affordable, shelf-stable, and tasty option for promoting gut health, with possible benefits for the gut barrier, antioxidants, and microbiome that require further research. Future research should explore its mechanisms, including oxidative stress, inflammation, and mucin gene regulation, and clinical or nutrigenomic efficacy. Increasing production and testing in various foods could position GBF as a nutraceutical in the functional food market. This work supports pregelatinized GBF's role in medical nutrition and gastrointestinal disease prevention.

## Supporting information

**S1 File. Raw data files.**
(XLSX)

## Author contributions

**Conceptualization:** Surasak Limsuwan, Sasitorn Chusri.

**Data curation:** Thammarat Kaewmanee, Pinanong Na-Phatthalung.

**Funding acquisition:** Surasak Limsuwan, Sasitorn Chusri.

**Investigation:** Thammarat Kaewmanee, Sineenart Sanpinit, Acharaporn Issuriya.

**Methodology:** Thammarat Kaewmanee, Sineenart Sanpinit, Acharaporn Issuriya.

**Supervision:** Thammarat Kaewmanee, Acharaporn Issuriya.

**Validation:** Samuel Abiodun Kehinde.

**Visualization:** Pinanong Na-Phatthalung, Samuel Abiodun Kehinde.

**Writing – original draft:** Surasak Limsuwan, Pinanong Na-Phatthalung, Sasitorn Chusri.

**Writing – review & editing:** Surasak Limsuwan, Sineenart Sanpinit, Pinanong Na-Phatthalung, Samuel Abiodun Kehinde, Sasitorn Chusri.

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
