## [Decision Letter · Decision Letter 0]

14 Oct 2025

Dear Dr. Chusri,

Thank you for submitting your manuscript to PLOS ONE. After careful consideration, we feel that it has merit but does not fully meet PLOS ONE’s publication criteria as it currently stands. Therefore, we invite you to submit a revised version of the manuscript that addresses the points raised during the review process.

We look forward to receiving your revised manuscript.

Kind regards,

José M. Alvarez-Suarez

Academic Editor

PLOS ONE

2. Please include your tables as part of your main manuscript and remove the individual files. Please note that supplementary tables (should remain/ be uploaded) as separate "supporting information" files

[The research received financial support from the National Science, Research, and Innovation Fund (NSRF) and Mae Fah Luang University (Fundamental Fund Grant No. 662A05032). It was also partly supported by the Southern Thailand Science Park (Collaborative Research Program; Co-Research-PsuCor-1/62), Prince of Songkla University. The manuscript's production was partially assisted through academic collaboration under the reinventing university system of Mae Fah Luang University.].

4. We note that your Data Availability Statement is currently as follows: [All relevant data are within the manuscript and its Supporting Information files]

6. Please include a copy of Table 1-4 which you refer to in your text on page10, 11 and 13.

Additional Editor Comments (if provided):

Reviewers' comments:

Reviewer's Responses to Questions

**Comments to the Author**

1. Is the manuscript technically sound, and do the data support the conclusions?

Reviewer #1: Yes

Reviewer #2: Partly

2. Has the statistical analysis been performed appropriately and rigorously?

Reviewer #1: Yes

Reviewer #2: No

3. Have the authors made all data underlying the findings in their manuscript fully available?

Reviewer #1: Yes

Reviewer #2: No

4. Is the manuscript presented in an intelligible fashion and written in standard English?

Reviewer #1: Yes

Reviewer #2: No

Reviewer #1: Overall assessment: This is a well-designed and comprehensive study that investigates the development of a functional food product with a clear therapeutic target. The manuscript is generally well-written, the methodology is sound, and the results support the conclusions. The work has significant potential for practical application in the functional food industry. However, there are areas where clarity, data presentation, and statistical rigor can be improved to strengthen the manuscript for publication.

A. Inconsistencies and points of confusion

1. Dosage Form in Animal Study: The animal study administers "flour" (GBF and PregelF3) at 1000 mg/kg. Administering a dry powder to rats via oral gavage is technically challenging and can cause aspiration. It is standard practice to suspend or dissolve the test substance in a vehicle (e.g., water, carboxymethyl cellulose). The manuscript must clarify how the flours were administered.

2. Statistical Analysis Presentation: While ANOVA is mentioned, the post-hoc test (Duncan's) is not always clearly applied in the results text. For example, in Table 3, some values are described as "significantly" different without the corresponding letters being referenced in the narrative. The figures (Fig 3) lack statistical annotation (e.g., letters or asterisks) to show which groups are significantly different from each other.

3. Sensory Evaluation Scale: The manuscript states a "9-point hedonic scale" is used but does not define the anchors (e.g., 1=dislike extremely, 9=like extremely). This is a minor but important detail for reproducibility.

4. Microbiological Data in Table 4: The table lists "ND" for Total Viable Count (TVC) at week 0 but `1.1x10^2` CFU/g at week 2. An increase from ND to 10² is unusual for a baked, low-moisture product and should be briefly explained (e.g., potential for post-baking contamination during handling, or clarification that "ND" means below a certain detection limit, like <10 CFU/g).

B. Section-by-section analysis with recommendations

ABSTRACT

Line 4-5: "well-recognized gluten-free flour" – This is true but not the primary focus. Consider rephrasing to emphasize its recognized gastroprotective potential in traditional medicine/preclinical studies.

Line 10-11: "systematically investigated" – Good phrasing, it accurately reflects the study's strength.

INTRODUCTION

Line 34-35: "However, in addition to this widespread ethnobotanical use, there is no evidence of its gastroprotective effects." – This statement is incorrect and a major error. The sentences immediately preceding this (Lines 30-33) cite references [13, 14, 15] that provide evidence of gastroprotective effects. This line must be deleted or heavily rephrased (e.g., "However, evidence for the specific cultivar Musa sapientum Linn., ABB group, cv. Kluai Namwa, used in Thai medicine is limited").

Line 41-42: "may also influence bioactivity" – This is a key point of your hypothesis. You can state this more confidently, e.g., "but its impact on bioactivity, particularly gastrointestinal protective potential, remains unexplored."

MATERIALS AND METHODS

Section 2.1:

Line 68-70: Specify the concentration of GBF in the slurry for each formulation (PregelF1-F4) here, or state that they are detailed in Table 1. It's currently in Table 1, but mentioning the range (20-50%) in the text improves flow.

Section 2.2:

Line 80-81: "using the published protocol" – This is too vague. You must cite the specific protocol or describe the method briefly (which you do in the following sentences). Delete "using the published protocol."

Section 2.4 (Gastroprotective effects):

Line 125-126: Crucial Omission. You must describe how the GBF and PregelF3 powders were prepared for oral administration to the rats. Were they suspended in water or another vehicle? This is a critical methodological detail.

Line 143: "Gastric contents were collected to measure pH" – Specify how the gastric contents were collected (e.g., by gentle scraping or washing with a fixed volume of water?).

RESULTS

Table 1:

The footnote says "Values are expressed as mean±SD (n=3)." This is good. Ensure all values in the table carry the same number of decimal places for consistency (e.g., Water Activity is given to 3 decimal places, which is fine, but be consistent).

Yield: It would be helpful to briefly speculate in the text on why yield decreases with higher GBF concentration (e.g., increased viscosity leading to more adhesion to the drums).

Table 2 & Sensory Evaluation:

The text states sPreF3 had the highest ratings, but the statistical analysis (letters) in Table 2 shows that for most attributes (Color, Flavor, Taste, Overall Appearance), sPreF3 is not significantly different from other formulations. The narrative should be tempered to reflect this, e.g., "sPreF3 received numerically higher ratings... and was significantly preferred for crunchiness over sPreF4."

Figure 2 & 3:

Figure 2: The labels (A-E) are missing from the figure itself. They must be added.

Figure 3: The graphs (A, B, C) must include asterisks () or letters to indicate significant differences between the groups. The description in the legend is not a substitute for visual statistical annotation on the graph.

Table 3 (Shelf Stability):

As mentioned under "Inconsistencies," the microbiological data needs a brief comment in the results or discussion (e.g., "The low but detectable TVC after week 2 suggests minor post-processing contamination, but counts remained stable and within acceptable limits throughout the study.").

The sensory scores show some significant fluctuations. This should be mentioned in the text, not just presented in the table.

DISCUSSION

Line 232-234: "This finding aligns with previous studies, which demonstrate that increased temperature during drying processes substantially boosts the swelling power and water solubility index of green bananas [33, 34]." – Excellent, situates your findings within existing literature.

Line 255-257: "Sensory evaluations indicated that snack bars containing 40% pregelatinized GBF (sPreF3) exhibited the highest acceptability..." – As per the comment on Results, adjust this statement to accurately reflect the statistical results from Table 2.

Line 270-271: "It should be highlighted that the pregelatinization process used in this study does not attenuate the gastroprotective properties of GBF." – This is a key conclusion and should be stated prominently.

Limitations Section (Line 295-301): This is well-written and strengthens the manuscript by showing scholarly awareness of the study's constraints.

CONCLUSION

Concise and accurately reflects the study's findings. No changes needed.

REFERENCES, FORMATTING, and TYPOGRAPHICAL ERRORS

Line 17 (Keywords): "peptic ulcers" is used here, but "gastric ulcers" is used in the title and throughout the text. For consistency and discoverability, consider using "gastric ulcers" as the keyword.

Line 108: "sPreF3" should be defined here when first introduced (e.g., "snack bar made with PregelF3 (sPreF3)").

General: Carefully proofread the manuscript for minor spacing and punctuation errors (e.g., "cm²" sometimes appears as "cm2").

This is a strong paper with a compelling story. The core experiments are valid, and the data supports the conclusions. The revisions required are primarily for clarity, statistical rigor, and accuracy rather than fundamental flaws in the research.

Priority Actions:

1. Correct the major error in the Introduction regarding existing evidence for gastroprotective effects.

2. Clarify the methodology for administering the flour in the animal study.

3. Add statistical annotations (letters or asterisks) to all figures and tables where comparisons are made.

4. Ensure the narrative in the Results and Discussion sections accurately reflects the statistical outcomes, particularly for the sensory data.

With these revisions, this manuscript will be significantly strengthened and should be considered for publication in a good journal.

Reviewer #2: The authors are advised to carefully address all the comments/suggestions/queries, I have made section-wise on the MS entitled "Formulation and In Vivo Evaluation of Unripe Banana Flour Snack Bars: A Promising Alternative for Medical Nutrition Therapy in Gastric Ulcers."

** Title

The title needs to be improved and more accurate.

** Abstract

- The aim is very confused and needs to be improved. I don't understand this statement: "Including functional foods that support gastrointestinal health in the diet can aid in preventing peptic ulcers and minimizing their recurrence," as it can't be the beginning of the abstract. Change it.

- Also, the aim needs to be rewritten to be better understood.

- The abstract is missing numerical data on key findings

- Keywords are general and vague. They need to be improved.

** Introduction

- The introduction is very short and scattered. It should cover all the points addressed in the study.

- Clear objective of the present work.

- The introduction should mention what is novel about this study.

*Methods

- All the physicochemical analyses need to be divided into subheading titles to be more accurate and clear as they are very confused.

- The sensory evaluation needs to be in a separate section.

- What did you mean by this title: Gastroprotective effects?

Moreover, this section is very confused, and the animal experiment needs to be organized as follows:

1. Experimental animals: Include animal details and ethical approval.

2. Laboratory diet composition: the authors need to mention a normal laboratory diet with references.

3. Study design: this section needs to be written in a scientific manner, as it is very confused, as the authors mentioned that there are two control groups: (1) sham control (distilled water); (2) vehicle control (distilled water); I don't know what the difference between them is, since both of them used distilled water. Also, the authors mentioned a positive control (omeprazole, 20 mg/kg). How is this positive control??? Omeprazole is a drug, and they used it to compare with the effect of using Unripe Banana Flour Snack Bars. Moreover, the positive control should be the group that had 80% ethanol (1 mL/rat).

4. Blood samples collection

5. Biochemical parameters: There are no biochemical parameters used in this study. The study should evaluate the activity of the treatment used in this study by measuring some biochemical parameters, including: Cytokines quantitation Tumor necrosis factor-α (TNF-α) and interleukins (IL-1, IL-6, IL-10, and IL-1β) should be measured

using ELISA kits. Moreover, lipid peroxidation, oxidative stress markers, enzymatic and non‑enzymatic antioxidant levels, malondialdehyde (MDA) concentration, nitric oxide (NO) concentration, glutathione peroxidase (GPx), glutathione S-transferase (GST), glutathione reductase (GR), superoxide dismutase (SOD), and reduced glutathione (GSH) should be measured.

6. Statistical analysis

- The authors need to compare all the treated groups to the positive and the drug control to prove their aim.

** Results

- Make the order of the results section the same as you will make in the methods section.

** Discussion needs to be improved and written in scientific words.

** The conclusion needs to be rewritten and more advanced.

** References

- Includes many old references and needs to be updated to be recent.

** There are many grammatical errors, and the writing level is low and needs to be improved.

- The authors need to organize this paper in a scientific manner and present their novelty.

.

Reviewer #1: **Yes:** Dr. Mohammad Nazrul Islam BhuiyanDr. Mohammad Nazrul Islam BhuiyanDr. Mohammad Nazrul Islam BhuiyanDr. Mohammad Nazrul Islam Bhuiyan

Reviewer #2: **Yes:** Dr. Dina Mostafa MohammedDr. Dina Mostafa MohammedDr. Dina Mostafa MohammedDr. Dina Mostafa Mohammed

---

## [Author Response · Author response to Decision Letter 1]

29 Dec 2025

The comment from the journal

Address the journal’s comment: Thank you very much for the suggestion. The author has proofread the revised manuscript and confirms that it meets PLOS ONE's style requirements, as mentioned.

2. Please include your tables as part of your main manuscript and remove the individual files. Please note that supplementary tables (should remain/ be uploaded) as separate "supporting information" files

Address the journal’s comment: Thank you for pointing this out for us. The author has included the tables in the revised manuscript, as requested.

3. Thank you for stating the following financial disclosure: [The research received financial support from the National Science, Research, and Innovation Fund (NSRF) and Mae Fah Luang University (Fundamental Fund Grant No. 662A05032). It was also partly supported by the Southern Thailand Science Park (Collaborative Research Program; Co-Research-PsuCor-1/62), Prince of Songkla University. The manuscript's production was partially assisted through academic collaboration under the reinventing university system of Mae Fah Luang University.]. Please state what role the funders took in the study. If the funders had no role, please state: "The funders had no role in study design, data collection and analysis, decision to publish, or preparation of the manuscript." If this statement is not correct you must amend it as needed.

Address the journal’s comment: Thank you for your kind concerns. We would like to declare the roles of each financial support as follows. The materials testing in this manuscript has been developed and supported by the research that received financial support from the National Science, Research, and Innovation Fund (NSRF) and Mae Fah Luang University (Fundamental Fund Grant No. 662A05032). The animal experiment was supported by the Southern Thailand Science Park (Collaborative Research Program; Co-Research-PsuCor-1/62), Prince of Songkla University. The manuscript's production was partially supported through academic collaboration under the reinventing university system at Mae Fah Luang University, which helped the international team work together. Furthermore, the article processing charge (APC) is supported by Mae Fah Luang University. Therefore, we would like to revise the funding statement as follows.

[The research received financial support from the National Science, Research, and Innovation Fund (NSRF) and Mae Fah Luang University (Fundamental Fund Grant No. 662A05032). It was also partly supported by the Southern Thailand Science Park (Collaborative Research Program; Co-Research-PsuCor-1/62), Prince of Songkla University. The manuscript's production was partially assisted through academic collaboration under the reinventing university system of Mae Fah Luang University. The article processing charge (APC) is supported by Mae Fah Luang University.]

4. We note that your Data Availability Statement is currently as follows: [All relevant data are within the manuscript and its Supporting Information files]

Authors do not need to submit their entire data set if only a portion of the data was used in the reported study. If your submission does not contain these data, please either upload them as Supporting Information files or deposit them to a stable, public repository and provide us with the relevant URLs, DOIs, or accession numbers. For a list of recommended repositories, please see https://journals.plos.org/plosone/s/recommended-repositories.

Address the journal’s comment: The information including the raw data used in this manuscript was provided in the Supporting Information files under the name ‘Final Report’.

Address the journal’s comment: The caption for each figure was included in the revised manuscript.

6. Please include a copy of Table 1-4 which you refer to in your text on page10, 11 and 13.

Address the journal’s comment: The Table 1-4 was included in the revised manuscript.

Address the journal’s comment: Thank you very much for your concerns, there is no recommendation to cite specific previously published works from the reviewers.

Reviewer #1: Overall assessment: This is a well-designed and comprehensive study that investigates the development of a functional food product with a clear therapeutic target. The manuscript is generally well-written, the methodology is sound, and the results support the conclusions. The work has significant potential for practical application in the functional food industry. However, there are areas where clarity, data presentation, and statistical rigor can be improved to strengthen the manuscript for publication.

A. Inconsistencies and points of confusion

1. Dosage Form in Animal Study: The animal study administers "flour" (GBF and PregelF3) at 1000 mg/kg. Administering a dry powder to rats via oral gavage is technically challenging and can cause aspiration. It is standard practice to suspend or dissolve the test substance in a vehicle (e.g., water, carboxymethyl cellulose). The manuscript must clarify how the flours were administered.

Address the reviewer’s comment:

Thank you for highlighting this important methodological detail. We agree that directly administering dry powder is not practical for oral gavage and may pose a risk to the animals. We have now clarified the administration procedure in the Materials and Methods (Section 2.4) as follows:

“Both GBF and PregelF3 powders were freshly suspended in distilled water immediately before dosing to obtain a homogeneous slurry suitable for oral administration. The suspensions were administered to the rats by oral gavage at a dose of 10 mL/kg body weight to ensure consistent delivery of the 1000 mg/kg dose. This method ensured accurate dosing, minimized the risk of aspiration, and complied with standard procedures for safe oral administration in rodents.”

This revision specifies the vehicle used, the preparation method, and ensures compliance with standard gavage safety practices. Thanks again for emphasizing this point and for helping to clarify our methodology.

2. Statistical Analysis Presentation: While ANOVA is mentioned, the post-hoc test (Duncan's) is not always clearly applied in the results text. For example, in Table 3, some values are described as "significantly" different without the corresponding letters being referenced in the narrative. The figures (Fig. 3) lack statistical annotation (e.g., letters or asterisks) to show which groups are significantly different from each other.

Address the reviewer’s comment: Thank you very much for your feedback. All tables and figure legends now reference consistent superscript letters. Figure 3 now includes asterisks and pound signs indicating statistical differences (* vs. vehicle; # vs. sham), and as a result, the text in Results was revised for accuracy.

3. Sensory Evaluation Scale: The manuscript states a "9-point hedonic scale" is used but does not define the anchors (e.g., 1=dislike extremely, 9=like extremely). This is a minor but important detail for reproducibility.

Address the reviewer’s comment: The information was added in the Methods section as follows.

“The 9-point hedonic scale was anchored as follows: 1 = dislike extremely, 2 = dislike very much, 3 = dislike moderately, 4 = dislike slightly, 5 = neither like nor dislike, 6 = like slightly, 7 = like moderately, 8 = like very much, and 9 = like extremely. ”

4. Microbiological Data in Table 4: The table lists "ND" for Total Viable Count (TVC) at week 0 but `1.1x10^2` CFU/g at week 2. An increase from ND to 10² is unusual for a baked, low-moisture product and should be briefly explained (e.g., potential for post-baking contamination during handling, or clarification that "ND" means below a certain detection limit, like <10 CFU/g).

Address the reviewer’s comment: Thank you very much, we added clarification in the Results and Table 3:

An acceptable microbiological quality was also achieved in the formulated snack bar, with total viable microbial counts stabilizing around 102 CFU/g after the second week. In comparison, yeast and mold counts remained generally low (<10 CFU/g). At week 0, total viable counts were below the detection limit (<10 CFU/g). The slight increase to approximately 10² CFU/g observed by week 2 may be attributed to minor post-baking contamination during handling or packaging.

B. Section-by-section analysis with recommendations

ABSTRACT

Line 4-5: "well-recognized gluten-free flour" – This is true, but not the primary focus. Consider rephrasing to emphasize its recognized gastroprotective potential in traditional medicine/preclinical studies.

Line 10-11: "systematically investigated" – Good phrasing, it accurately reflects the study's strength.

Address the reviewer’s comment: Thank you very much for your kind suggestion and encouraging our research. The abstract was revised by rephrasing the opening sentence, adding a clear objective statement, and including numerical results. Additionally, the keywords have been updated to be more specific: gastroprotection; pregelatinized banana flour; resistant starch; functional food; ethanol-induced ulcer model; snack bar formulation.

INTRODUCTION

Line 34-35: "However, in addition to this widespread ethnobotanical use, there is no evidence of its gastroprotective effects." – This statement is incorrect and a major error. The sentences immediately preceding this (Lines 30-33) cite references [13, 14, 15] that provide evidence of gastroprotective effects. This line must be deleted or heavily rephrased (e.g., "However, evidence for the specific cultivar Musa sapientum Linn., ABB group, cv. Kluai Namwa, used in Thai medicine is limited").

Line 41-42: "may also influence bioactivity" – This is a key point of your hypothesis. You can state this more confidently, e.g., "but its impact on bioactivity, particularly gastrointestinal protective potential, remains unexplored."

Address the reviewer’s comment: Thank you very much for the very informative suggestion. The introduction section was revised to remove the incorrect claim that no gastroprotective evidence exists, to indicate the gap in knowledge, and to strengthen the rationale and novelty. The following information was added.

“Among them, the unripe Musa species used to make green banana flour (GBF) can be singled out as a highly nutritious source of resistant starch, phenolic antioxidants, and non-digestible fiber [8-9]. Its gastroprotective properties have been demonstrated in several preclinical studies, which show reductions in gastric and intestinal inflammation across various models [10-15]. Moreover, its long-range ethnopharmacological history of use in traditional Thai and Sudanese medicine for the treatment of peptic ulcer disease has long been regarded as evidence of its therapeutic potential [13,16].

Nonetheless, there are challenges associated with the direct application of native GBF of food formulations. Its low solubility, roughness, and low digestibility limit its sensory properties and industrial application [17,18].”…

“Although these advantages are well known, a research gap remains regarding the effects of pregelatinization on the gastroprotective activity of GBF and its functional properties in consumer-acceptable edible foods. The study was thus aimed at determining the physicochemical properties of pregelatinized drum-dried GBF, comparing its gastroprotective effects with those of native GBF using an ethanol-induced gastric ulcer rat model, and evaluating shelf-stable snack bars containing pregelatinized GBF in terms of their sensory and storage properties.

We posit that pregelatinization does not alter the gastroprotective performance of GBF but improves its functional and physicochemical characteristics, thereby developing a palatable, stable, health-promoting snack bar that can be used in gastrointestinal nutritional support.”

MATERIALS AND METHODS

Section 2.1: Line 68-70: Specify the concentration of GBF in the slurry for each formulation (PregelF1-F4) here, or state that they are detailed in Table 1. It's currently in Table 1, but mentioning the range (20-50%) in the text improves flow.

Address the reviewer’s comment: The concentration of GBF was specified as recommended in Section 2.1.

‘The study employed a completely randomized design (CRD) to investigate the effects of GBF concentrations (20%, 30%, 40%, and 50% w/v) on the physicochemical properties of drum-dried formulations designated PregelF1, PregelF2, PregelF3, and PregelF4, respectively.’

Section 2.2: Line 80-81: "using the published protocol" – This is too vague. You must cite the specific protocol or describe the method briefly (which you do in the following sentences). Delete "using the published protocol."

Address the reviewer’s comment: The text was deleted as suggested. The specific protocol was cited as mentioned.

Section 2.4 (Gastroprotective effects): Line 125-126: Crucial Omission. You must describe how the GBF and PregelF3 powders were prepared for oral administration to the rats. Were they suspended in water or another vehicle? This is a critical methodological detail.

Address the reviewer’s comment: The information has been added to Section 2.6, Gastroprotective effects, as follows.

“Both GBF and PregelF3 powders were freshly suspended in distilled water immediately before dosing to obtain a homogeneous slurry suitable for oral administration. The suspensions were administered to the rats by oral gavage at a dose of 10 mL/kg body weight to ensure consistent delivery of the 1000 mg/kg dose. This method ensured accurate dosing, minimized the risk of aspiration, and complied with standard procedures for safe oral administration in rodents.”

Line 143: "Gastric contents were collected to measure pH" – Specify how the gastric contents were collected (e.g., by gentle scraping or washing with a fixed volume of water?).

Address the reviewer’s comment: The section has been revised for greater clarity, as follows.

“After the stomachs were excised and opened along the greater curvature, the gastric contents were gently collected by rinsing the stomach lumen with a fixed volume of distilled water (2 mL) using

---

## [Decision Letter · Decision Letter 1]

10 Mar 2026

Dear Dr. Chusri,

Thank you for submitting your manuscript to PLOS ONE. After careful consideration, we feel that it has merit but does not fully meet PLOS ONE’s publication criteria as it currently stands. Therefore, we invite you to submit a revised version of the manuscript that addresses the points raised during the review process.

We look forward to receiving your revised manuscript.

Kind regards,

José M. Alvarez-Suarez

Academic Editor

PLOS One

Journal Requirements:

Reviewers' comments:

Reviewer's Responses to Questions

**Comments to the Author**

Reviewer #1: All comments have been addressed

Reviewer #2: All comments have been addressed

2. Is the manuscript technically sound, and do the data support the conclusions?

Reviewer #1: Yes

Reviewer #2: Yes

3. Has the statistical analysis been performed appropriately and rigorously?

Reviewer #1: Yes

Reviewer #2: Yes

4. Have the authors made all data underlying the findings in their manuscript fully available?

Reviewer #1: Yes

Reviewer #2: Yes

5. Is the manuscript presented in an intelligible fashion and written in standard English?

Reviewer #1: Yes

Reviewer #2: Yes

Reviewer #1: Review Report: Revised Manuscript PONE-D-25-49266R1

Title: Pregelatinized Green Banana Flour Snack Bars: Physicochemical Properties, Sensory Quality, and Gastroprotective Potential in an Ethanol-Induced Ulcer Model

General Assessment: The authors have submitted a thoroughly revised manuscript that demonstrates significant improvement in response to the reviewers' comments. The study is well-designed, investigating a novel functional food application with clear therapeutic relevance. The revisions have enhanced the manuscript's clarity, methodological rigor, and statistical presentation. The core findings—that drum-drying pregelatinization improves the functional properties of green banana flour (GBF) without diminishing its gastroprotective efficacy—are supported by the data and are of potential interest to the fields of food science and nutritional therapy.

Assessment of Author Responses to Reviewers:

The authors have provided a detailed point-by-point response and have made substantial changes to the manuscript. A review of the revised document confirms that the major concerns of both reviewers have been addressed adequately.

1. Reviewer #1's Priority Actions:

Major Error in Introduction: Corrected. The incorrect statement about a lack of gastroprotective evidence has been removed, and the introduction now accurately cites existing literature while clearly stating the study's novel focus on pregelatinization.

Animal Study Methodology: Clarified. The Methods section (2.6) now explicitly states that GBF and PregelF3 were suspended in distilled water for oral gavage, addressing safety and practicality concerns.

Statistical Annotations: Added. Figure 3 now includes asterisks and pound signs to denote statistical significance. Table footnotes and in-text references to superscript letters are consistent.

Narrative Alignment with Statistics: Adjusted. The text in the Results (3.2) and Discussion regarding sensory evaluation now accurately reflects the statistical outcomes from Table 2, tempering claims where differences were not significant.

2. Reviewer #2's Major Concerns:

Title, Abstract, and Keywords: Revised satisfactorily. The title is more accurate, the abstract is clearer and includes key numerical data, and keywords are more specific.

Introduction and Novelty: Significantly strengthened. The introduction is more comprehensive, clearly outlines the rationale, states clear objectives, and highlights the novelty of investigating pregelatinization's impact on both functionality and bioactivity.

Methods Reorganization: Improved. The Methods section is now better organized with subheadings. The animal experiment description is more detailed, including diet composition and a clearer explanation of control groups.

Study Limitations: Acknowledged. The authors have appropriately noted the absence of blood-based biochemical markers (e.g., cytokines, oxidative stress parameters) as a limitation, explaining it was due to practical constraints while justifying the sufficiency of their chosen endpoints.

Remaining Issues and Suggestions for Minor Revision:

While the manuscript is much improved, some minor issues require correction before final acceptance:

1. Grammar and Language: While greatly improved, some minor grammatical inconsistencies persist.

Line 44: "of food formulations" should be "in food formulations".

Line 126: "pregelatinized GBF (pregelF1..." should be "pregelatinized GBF (PregelF1..." for consistency with defined terms.

Line 192: "...confirmed the model's sensitivity." Consider "confirmed the efficacy of the positive control and the model's sensitivity."

Line 271: "Pregelatinized GBF developed in this present work serves..." -> "The pregelatinized GBF developed in this study serves..."

Conclusion Section: The conclusion is comprehensive but slightly verbose. Consider tightening some sentences for conciseness (e.g., the sentence beginning "In addition to its proven antiulcer activity..." could be split or streamlined).

2. Figure and Table Consistency:

Table 1: The decimal places are now consistent, which is good.

Figure 2: Labels (A-E) are present. Confirm that these are clearly visible in the high-resolution figure files uploaded to the submission system.

Table 3/4 Numbering: The manuscript text refers to a "Table 3" for shelf stability data and a "Table 4" for in vivo effects. However, in the provided manuscript body, the in vivo table appears as Table 4 and the shelf-stability table as Table 3. The authors must ensure the numbering sequence in the main text (Tables 1, 2, 3, 4) is correct and sequential based on their first mention.

3. Funding Statement (Lines 380-384): The updated statement in the manuscript is clear and includes the required disclaimer ("The funders had no role...").

• 4. Refere4nces: Please check all references and include all latest references within last 5 yeasr like as: https://doi.org/10.1016/j.afres.2024.100431; https://doi.org/10.1371/journal.pone.0307909 etc

Upon satisfactory resolution of these minor points, this manuscript will be ready for publication.

Reviewer #2: (No Response)

.

Reviewer #1: **Yes:** Dr. Mohammad Nazrul Islam BhuiyanDr. Mohammad Nazrul Islam BhuiyanDr. Mohammad Nazrul Islam BhuiyanDr. Mohammad Nazrul Islam Bhuiyan

Reviewer #2: **Yes:** Assoc. Prof. Dina Mostafa MohammedAssoc. Prof. Dina Mostafa MohammedAssoc. Prof. Dina Mostafa MohammedAssoc. Prof. Dina Mostafa Mohammed

---

## [Author Response · Author response to Decision Letter 2]

17 Mar 2026

We would like to express our sincere gratitude to the Editor and reviewers for their careful evaluation of our manuscript entitled “Pregelatinized Green Banana Flour Snack Bars: Physicochemical Properties, Sensory Quality, and Gastroprotective Potential in an Ethanol-Induced Ulcer Model.” The reviewers’ insightful comments and suggestions have been invaluable in improving the clarity, consistency, and overall quality of the manuscript.

All comments were carefully considered and addressed in the revised manuscript. Revisions have been made accordingly and are indicated in the manuscript. A detailed point-by-point response to each comment is provided below.

Reviewer Comments and Author Responses

1. Grammar and Language

Comment:

Line 44: “of food formulations” should be “in food formulations”.

Response:

We thank the reviewer for pointing out this language correction. The preposition has been revised to improve grammatical accuracy.

Comment:

Line 126: “pregelatinized GBF (pregelF1…” should be “pregelatinized GBF (PregelF1…”.

Response:

We appreciate the reviewer’s attention to detail. The capitalization of the abbreviation has been corrected to maintain consistency with the defined terminology used throughout the manuscript.

Comment:

Line 192: “…confirmed the model's sensitivity.” Consider revising to “confirmed the efficacy of the positive control and the model's sensitivity.”

Response:

Thank you for this helpful suggestion. The sentence has been revised to clarify that the results demonstrate both the effectiveness of the positive control treatment and the sensitivity of the experimental ulcer model.

Comment:

Line 271: “Pregelatinized GBF developed in this present work serves…” should be revised to “The pregelatinized GBF developed in this study serves…”.

Response:

We appreciate this suggestion to improve the academic flow of the sentence. The wording has been revised accordingly to enhance clarity and readability.

Comment:

Conclusion section: The conclusion is comprehensive but slightly verbose. Consider streamlining the sentences.

Response:

We thank the reviewer for this valuable suggestion. The conclusion section has been revised to improve conciseness and readability. Long sentences were shortened and restructured, particularly those describing the anti-ulcer activity of the developed snack bar formulation.

2. Figure and Table Consistency

Comment:

Table 1: The decimal places are now consistent.

Response:

We confirm that the decimal formatting in Table 1 has been carefully checked and maintained consistently throughout the revised manuscript.

Comment:

Figure 2: Confirm that labels (A–E) are clearly visible in the high-resolution files.

Response:

We confirm that the labels (A–E) are clearly visible in the high-resolution figure files submitted with the revised manuscript.

Comment:

Table 3/4 Numbering: Ensure the numbering sequence is correct based on the first mention in the text.

Response:

We appreciate the reviewer’s attention to formatting details. The numbering sequence of Tables 3 and 4 has been carefully rechecked and corrected to ensure that tables appear sequentially according to their first citation in the manuscript.

3. Funding Statement

Comment:

Ensure that the funding statement includes the required disclaimer.

Response:

We confirm that the funding statement already includes the mandatory disclaimer required by the journal. The section was reviewed to ensure compliance with journal guidelines and remains unchanged.

4. References

Comment:

Please check all references and include recent citations from the last five years.

Response:

Thank you for this suggestion. The reference list has been carefully reviewed and updated to include recent and relevant literature published within the last five years. In particular, the recommended articles (DOIs: 10.1016/j.afres.2024.100431 and 10.1371/journal.pone.0307909) have been incorporated where appropriate. All references were also checked for accuracy and consistency in formatting.

---

## [Decision Letter · Decision Letter 2]

8 Apr 2026

Pregelatinized Green Banana Flour Snack Bars: Physicochemical Properties, Sensory Quality, and Gastroprotective Potential in an Ethanol-Induced Ulcer Model

PONE-D-25-49266R2

Dear Dr. Chusri,

We’re pleased to inform you that your manuscript has been judged scientifically suitable for publication and will be formally accepted for publication once it meets all outstanding technical requirements.

Kind regards,

José M. Alvarez-Suarez

Academic Editor

PLOS One

Additional Editor Comments (optional):

Reviewers' comments:

Reviewer's Responses to Questions

**Comments to the Author**

Reviewer #1: All comments have been addressed

2. Is the manuscript technically sound, and do the data support the conclusions?

Reviewer #1: Yes

3. Has the statistical analysis been performed appropriately and rigorously?

Reviewer #1: Yes

4. Have the authors made all data underlying the findings in their manuscript fully available?

Reviewer #1: Yes

5. Is the manuscript presented in an intelligible fashion and written in standard English?

Reviewer #1: Yes

Reviewer #1: The authors have addressed all the issues and made the necessary corrections. The revised manuscript appears suitable for acceptance and publication.

.

Reviewer #1: **Yes:** Dr. Mohammad Nazrul Islam BhuiyanDr. Mohammad Nazrul Islam BhuiyanDr. Mohammad Nazrul Islam BhuiyanDr. Mohammad Nazrul Islam Bhuiyan

---

## [Editor Report · Acceptance letter]

PONE-D-25-49266R2

PLOS One

Dear Dr. Chusri,

I'm pleased to inform you that your manuscript has been deemed suitable for publication in PLOS One. Congratulations! Your manuscript is now being handed over to our production team.

Kind regards,

on behalf of

Professor José M. Alvarez-Suarez

Academic Editor

PLOS One